# A Successful Dental Care Referral Program for Low-Income Pregnant Women in New York

**DOI:** 10.3390/ijerph182312724

**Published:** 2021-12-02

**Authors:** Stefanie L. Russell, Steven J. Kerpen, Jill M. Rabin, Ronald P. Burakoff, Chengwu Yang, Shulamite S. Huang

**Affiliations:** 1Department of Epidemiology and Health Promotion, New York University College of Dentistry, New York, NY 10012, USA; shulamite.huang@nyu.edu; 2Northwell Health Long Island Jewish Medical Center, New Hyde Park, NY 11040, USA; SKerpen@Northwell.edu (S.J.K.); JRabin@northwell.edu (J.M.R.); RBurakoff@Northwell.edu (R.P.B.); 3Department of Population and Quantitative Health Sciences, UMass Chan Medical School, Worcester, MA 01605, USA; Chengwu.Yang@umassmed.edu

**Keywords:** pregnancy, dental care, prenatal care, low-income women, racial–ethnic minority women

## Abstract

Despite evidence-based guidelines that advocate for dental care during pregnancy, dental utilization among pregnant women remains low, especially among low-income and racial–ethnic minority women. We investigated self-reported dental care referral and self-reported dental care attendance among a group of 298 low-income, largely racial–ethnic minority pregnant women attending two suburban prenatal care clinics that had integrated dental care referrals into their prenatal care according to these guidelines. We administered a questionnaire that asked women: (1) whether they had been referred for care by their prenatal care provider; (2) whether they had been seen by a dentist during pregnancy. Among those women who were eligible for a dental care referral (those who reported having dental symptoms, and those not having a recent dental visit), we found that 73.0% reported that they had indeed been referred for dental care by their prenatal provider, while the remaining women reported either no referral (23.5%, *n* = 67) or were not sure whether they had been referred (3.5%, *n* = 10). Among those who reported a dental care referral, 67.3% (*n* = 140) reported that they saw a dentist during their pregnancy, while of those who reported no dental care referral only 35.1% (*n* = 27) reported a dental visit (Chi-Sq. = 24.1, df = 1, *p* < 0.001). Having received a dental referral was a significant predictor of reporting a dental visit during pregnancy, with women who received a referral being 4.6 times more likely to report a dental visit during pregnancy compared to those women who did not report a referral. These results demonstrate that vulnerable pregnant women referred for dental care by their prenatal provider will indeed seek and utilize dental care when offered. This dental referral program may serve as a model for improving the utilization of dental care among this population.

## 1. Introduction

An adage says that “for every child, a woman loses a tooth.” [1]. It has long been recognized that during pregnancy women are at an increased risk of oral disease primarily because of hormonal changes that adversely affect the oral soft tissues [2]. Gingivitis, inflammation of the gingiva, occurs in most pregnant women and worsens as pregnancy progresses [3,4]. Periodontal disease, characterized by destruction of the tooth-supporting structures (connective tissue and bone), is common in pregnant women [5] and can progress during pregnancy, especially if not treated [6]. Women may also be at increased risk for caries (which causes pain, infection, and tooth loss) during pregnancy due to pregnancy-related changes in diet, Ref. [7] salivary changes, Ref. [8] or reduction/weakening of tooth enamel resulting from hyperemesis gravidarum (“morning sickness”) [2]. Low-income and minority women are at increased risk of dental disease, and disparities in oral health are evident in both pregnant and non-pregnant women of child-bearing age [5]. A recent study that used NHANES data found that 87.2% of pregnant women had experienced dental caries, and 28.3% had untreated dental caries. Untreated caries was more than twice as common among non-Hispanic Black (45.3%) and Mexican American (42.2%) women compared with non-Hispanic White women (17.8%). More than half of pregnant women (53.0%) living below the Federal Poverty Level had untreated caries. Periodontal disease, which ordinarily is uncommon among persons less than 35 years old, affected 9.3% of Mexican American women and 5.7% of women with low education [5].

In 2006, the New York State Department of Health published a set of evidence-based guidelines that advocate for the provision of preventive dental care and timely treatment of dental disease during pregnancy [9]. Despite these and other guidelines [10], and the overwhelming evidence that dental care during pregnancy is safe and effective [11,12], utilization of dental services among pregnant women has remained low, especially among low-income women [13,14,15,16]. To date, there are few reports of successful evidence-based programs designed to improve dental care utilization among pregnant women.

In 2007, a program established by a private practice periodontist (SK) and an obstetrician (JR) in suburban New York sought to improve oral health of pregnant women by following the recommendations from the 2006 New York State Guidelines for Oral Health Care for Pregnant Women [17]. Women seen for prenatal care in the clinics of Long Island Jewish Hospital in New Hyde Park and North Shore Hospital in Manhasset, Long Island (now both part of the Northwell Health System) were routinely referred for dental care using an algorithm from the New York State Guidelines (Figure 1). Women seen in these facilities are low-income and are covered by Medicaid during pregnancy. Women who reported: (1) having a dental problem (including “bleeding gums, toothache, cavities, loose teeth, teeth that do not look right or other problems”); (2) not having seen a dental provider in the previous six months were referred by their prenatal care provider to a dentist. Women who reported not having their own dentist were offered an opportunity to be scheduled with a local dentist who accepted Medicaid insurance as payment.

Investigators not associated with the prenatal oral health program from the New York University College of Dentistry (SR, CY, SH) evaluated the effectiveness of the program by surveying women from the prenatal clinics regarding dental care referral and dental care utilization during pregnancy. Our objectives were: (1) to evaluate whether women reported being referred for needing dental evaluation, prevention and care; (2) to determine whether those who reported referral were more likely to report having seen a dentist during pregnancy compared to those who did not report a referral.

## 2. Materials and Methods

This study was a cross-sectional analytic study. Over a two-month period in the summer of 2010, we recruited Medicaid-eligible pregnant women, aged 18+, from the two targeted prenatal clinics and asked them to complete written questionnaire in English or Spanish. Participants received a USD 5 gift card for completing the questionnaire. All surveys were anonymous and voluntary. This survey derived from a similar questionnaire that was used previously in North Carolina [18].Survey questions recorded demographics, pregnancy characteristics, perceived oral health and oral health behaviors, dental symptoms during pregnancy and dental care history, dental care referral during pregnancy and having a dental visit during pregnancy.

All survey data were entered into a database, verified and cleaned and analyzed using SPSS v.20. We report means and proportions and compared groups regarding dental care utilization during pregnancy (reporting a dental visit during their current pregnancy) using Chi-Square tests. We controlled for identified confounding variables using logistic regression. This study was approved by the New York University Medical Center and the Long Island Jewish/North Shore Institutional Review Boards.

## 3. Results

### 3.1. Descriptive Analyses

**Demographics and Pregnancy Characteristics:** We approached 317 women, and 298 (94.0%) women volunteered to participate and completed the survey. We gathered data from a total of 298 women who we approached in the two prenatal clinics (66.1% of the sample was from the LIJ facility while 33.9% was from the Northshore facility). In summary (see Table 1), the women were largely racial–ethnic minorities (Black = 29.4%, Hispanic = 39.1%, Asian = 12.8%). Roughly half were US immigrants (47.2%). Of the women who reported being born outside of the US (*n* = 137), most came from Central America (28.5%), the Caribbean (24.1%), South America (16.8%) and South Asia (16.8%). Most of the women (62.5%) reported completing high school and/or some college, and almost one in five (19.2%) were college graduates. Most women (92.9%) reported a family income of under 20,000 USD/year. Most women (77.7%) were married or living with a partner. The majority of the surveys (87.2%) were conducted in English. Women came from the surrounding areas of suburban Long Island (41.7%) and the neighboring borough of Queens, New York City (51.3%). This was the first pregnancy for 40.2% of the women, and more than half (53.4%) had not previously given birth. The majority of women reported initiating prenatal care during their first trimester of pregnancy (77.5%), and we surveyed most women (49.0%) when they were in their second trimester. Most women had symptoms of pregnancy including nausea (71.8%), vomiting (56.7%) and fatigue (68.8%). Few women reported gestational diabetes (5.0%) or hypertension (2.3%) during pregnancy.

**Perceived Oral Health/Dental Care Characteristics:** Oral health and dental care characteristics of the sample are shown in Table 2.

The majority of women rated their oral health as either excellent/very good (27.1%) or good (46.3%). More than half of the women reported brushing their teeth twice daily (55.7%), but only about one quarter (24.2%) reported cleaning between their teeth daily. The majority reported not having any missing teeth (63.4%), and few (10.4%) had ever been told that they had gum disease. More than half of the women (60.7%) reported seeing a dentist at least once yearly, and 40.3% reported seeing a private dentist (as opposed to going to a public or hospital dental clinic) for routine dental care when not pregnant. Two of five women (42.0%) reported that finding a dentist was somewhat or very difficult for them, and about half (54.0%) reported using Medicaid to pay for dental care when not pregnant. Women reported substantial oral symptoms during their current pregnancy including bleeding gums (47.7%), dental pain (44.1%), swollen gums (31.2%), having a cavity (28.5%) and having a loose tooth or teeth (10.1%).

**Dental Care Referral and Dental Visit during Current Pregnancy:** Almost ¾ of the women (72.5%) reported that they had been referred to a dentist during their current pregnancy by their prenatal care provider or clinic. More than half of the women surveyed (56.0%) reported that they had seen a dentist during the current pregnancy.

### 3.2. Bivariate Analyses

We evaluated demographic and pregnancy characteristics that were associated with oral health and dental care during pregnancy. White women were more likely than minority women to report a dental visit during pregnancy (75.0% for White vs. 51.7% of Black, 56.5% of Hispanic, 59.0% Asian; Chi-sq. 4.5, *p* = 0.03). We found that age, place of birth (US vs. non-US), education, income, marital status (not married vs. married), place of residence or language were not related to reporting a dental visit during pregnancy. We also found that increased frequency of dental care when not pregnant was associated with reporting a dental visit during the current pregnancy (Chi-sq. 15.7, df = 4, *p* = 0.003), as was having swollen gums during pregnancy (76.3% vs. 46.6%; Chi-sq. 23.0, df = 1, *p* ≤ 0.001). Other dental variables (perceived oral health, missing teeth, having gum disease, brushing/interproximal cleaning, payment method for dental care) were not associated with a dental visit during the current pregnancy.

When we applied the NY State Oral Health in Pregnancy Guidelines to our data, we found that 95.3% (*n* = 284) of the women we surveyed were eligible for a dental referral (we excluded those who reported having a recent dental visit only if they had no oral symptoms (4.7%, *n* = 13)). Of those women eligible for a dental referral, almost three quarters (73.0%, *n* = 208) reported that they had indeed been referred for dental care by their prenatal provider, while the remaining reported either no referral (23.5%, *n* = 67) or were not sure whether they had been referred (3.5%, *n* = 10). We classified those who were not sure whether they had been referred as not being referred in subsequent analyses. Among those who reported a referral, 67.3% (*n* = 140) reported that they saw a dentist during their pregnancy, while of those who reported no referral (or who were not sure whether they had received a referral), only 35.1% (*n* = 27) reported a dental visit (Chi-Sq. =24.1, df = 1, *p* < 0.001).

**Multivariate Analysis:** We further evaluated whether women who were referred for a dental visit by their prenatal care provider/clinic were more likely to report a dental visit controlling for race/ethnicity (White vs. minority), and found that, when controlling for minority racial–ethnic status, women who reported having received a referral for dental care by their prenatal provider/clinic were 4.6 times more likely to report a dental visit during pregnancy compared to those who reported no such referral.

## 4. Discussion

There are profound disparities in oral health among racial–ethnic and income groups in the US [19,20]. During pregnancy, women are at increased risk of oral disease due to hormonal changes and behavioral changes [2,3,4]. Unfortunately, those women most vulnerable to oral disease (racial–ethnic minorities, low-income women) have the lowest rates of dental utilization during pregnancy [13,14,15]. Importantly, for these vulnerable women, pregnancy may be the only time when they have coverage for dental care via the Medicaid program, as is the case in New York State [13]. National and state guidelines advocate for improving oral health by increasing dental care utilization during pregnancy and recommend that pregnant women be seen by a dentist for therapeutic or preventive treatment [9,10]. However, there have been few reports of programs designed to increase dental care utilization among vulnerable women.

Oral health guidelines for pregnant women published by New York State and others recommend that prenatal care providers integrate oral health into prenatal health by including screening questions likely to identify those women likely to require dental care. Several investigations have aimed to improve oral health of women during pregnancy by increasing dental care utilization. Cibulka et al. randomized a group of low-income, mostly African American pregnant women in the midwestern US to a nurse practitioner-directed oral care program that included a dental care referral or to a control where there was no such program. Following the intervention, 56.9% of the experimental group vs. 32.9% of the control group had seen a dentist during pregnancy [21]. In rural/suburban North Carolina, Jackson et al. instituted a Quality Improvement (QI) Project designed to improve access to oral health care for low-income pregnant women. This prospective cohort study found that 43% (55/126) of pregnant women referred for dental care attended a dental appointment [22]. A second report from this group reported an increase of 15.9% in dental visits among low-income pregnant women following their QI intervention compared to baseline [23]. A randomized clinical trial aimed at increasing dental attendance among pregnant women in Australia by oral health education and dental care referral by midwives found that the intervention was successful in increasing dental care utilization among by participants in the intervention group, where 87.2% of the women reported a dental visit compared to the control group where only 20.2% of the women reported a dental visit [24]. Finally, a study in Oregon that used motivational interviewing to improve oral health among a population of mostly White, low-income rural women found that 92% of the prenatal MI group had a dental visit, but also found that 94% in the control group reported a dental visit [25]. Aside from the final study where dental utilization was already very high, these studies collectively show that improvement of dental care utilization among low-income pregnant women is possible.

Our study evaluated data from an ongoing program that adopted use of oral health screening within the prenatal setting to increase dental care utilization among low-income women at high risk of disease. We found that routinely asking pregnant women whether they were having a dental problem or whether they had not seen a dental provider in the previous six months and using the referral form from the New York State Department of Health (Figure 2) resulted in a high rate of dental care referrals, and women who reported receiving a dental referral were more than four times more likely to report visiting a dentist during pregnancy compared to those without a referral. These results demonstrate that low-income pregnant women (a group with traditionally low rates of dental utilization but high unmet dental care needs) who are referred for dental care by their prenatal provider will indeed seek and utilize dental care when offered. Based on these findings, we believe that this dental referral program may serve as a model for improving access to and utilization of dental care by low-income pregnant women.

**Study limitations:** There are several limitations of the present study. Firstly, we collected data from a convenience sample of women who volunteered to complete surveys and are therefore susceptible to selection bias. Additionally, data were self-reported, and self-reported data regarding either referrals or dental visits during pregnancy were not validated via chart review, making our data susceptible to reporting bias. In addition, the data are cross-sectional in nature, so we have the ability to identify associations and not cause and effect. Finally, data were collected early in the life of the program, in 2010. However, the program is ongoing, and guidelines have not changed regarding referral protocols since the original *New York State Guidelines* document was published in 2006. Despite these limitations, we believe that these biases would have affected both groups (referred and non-referred) equally, and the large difference (4-fold) that we found in reported dental care utilization among women who reported a referral likely reflects a positive association of referrals and dental care utilization during pregnancy. In addition, we collected our data from women during pregnancy as opposed to following pregnancy, so some women who had been referred might not have yet visited a dentist as recommended by their prenatal care provider. Our results may not be generalizable to populations of women who are limited in their ability to access dental care. The New York State Medicaid program offers comprehensive dental care (exams, cleanings, fillings, etc.) to participants, and this program benefited from having a nearby dental office that was available to see the women from the clinic [17,26].

## 5. Conclusions

We found that low-income pregnant women (a group with traditionally low rates of dental utilization but high unmet needs) who report referral for dental care by their prenatal provider also report dental care utilization more often during pregnancy compared to those who report no referral. Routinely integrating dental screening questions into prenatal care resulted in a high rate of reported dental referrals. Our findings are in line with other investigations that have found that women who reported receiving a dental referral are more likely to report visiting a dentist during pregnancy. Based on these findings, we believe that this dental referral program may serve as a model for improving access to and utilization of dental care by low-income pregnant women.

## Figures and Tables

**Figure 1 ijerph-18-12724-f001:**
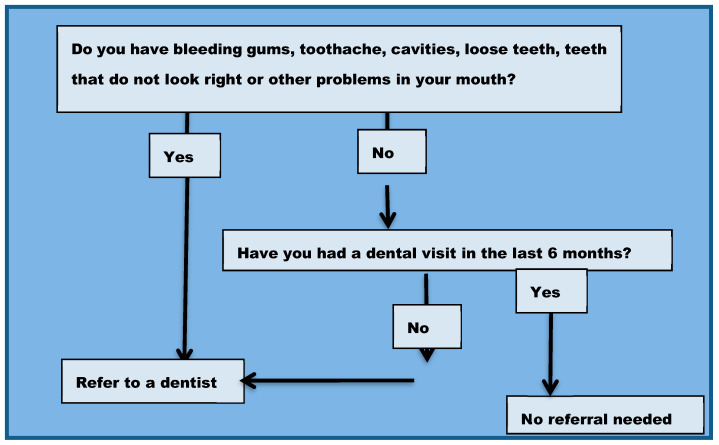
Dental referral algorithm.

**Figure 2 ijerph-18-12724-f002:**
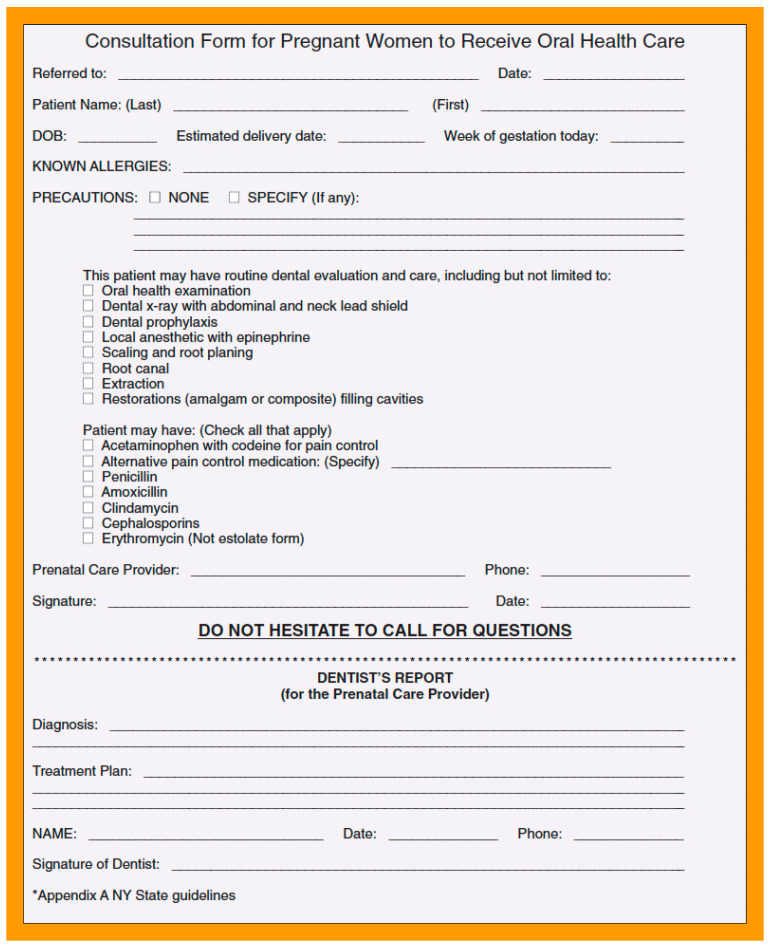
Dental referral form.

**Table 1 ijerph-18-12724-t001:** Demographic and pregnancy characteristics of the study participants.

Demographic Characteristics	Total (*n* = 298)
**Age (mean years ± ds)**	26.5 ± 5.4
**Race/ethnicity**	% (*n*)
White	10.7% (31)
Black	29.4% (86)
Asian	12.8% (37)
Hispanic	39.1% (113)
**Place of Birth**	
USA, including PR	52.5% (156)
**Education**	
<HS graduate	18.2% (53)
HS graduate/Some college	62.5% (182)
≥College graduate	19.2% (56)
**Family Income**	
≤10,000 USD/year	28.9% (86)
10,001–20,000 USD/year	33.2% (99)
≥20,000/year	30.8% (92)
**Marital Status**	
Married/living as married	77.7% (226)
**Language**	
English	87.2% (259)
Spanish	12.1% (36)
**Place of Residence**	
Long Island	41.7% (123)
Queens (NYC)	51.3% (153)
Brooklyn (NYC)	5.7% (17)
**Pregnancy Characteristics**	
**Gravidity** (# pregnancies)	
1	40.3% (120)
2	29.2% (87)
≥3	30.5% (91)
**Parity** (# live births)	
0	53.4% (159)
1	23.8% (71)
≥2	22.5% (67)
**First Prenatal Visit**	
first trimester	77.5% (231)
second trimester	30.8% (92)
third trimester	2.4% (7)
**Trimester at Time of Survey**	
1	12.4% (37)
2	49.0% (146)
3	38.3% (114)
**Pregnancy Symptoms**	
nausea	71.8% (214)
vomiting	56.7% (169)
fatigue	68.8% (205)
**Pregnancy Complications**	
gestational diabetes	5.0% (15)
hypertension	2.3% (7)

**Table 2 ijerph-18-12724-t002:** Reported oral health/dental care characteristics of the study participants.

	Total (*n* = 298) % (*n*)
**Perceived Oral Health**	
excellent/very good	27.1% (78)
good	46.3% (138)
fair/poor	24.2% (72)
**Brushing Frequency**	
≤once daily	4.3% (13)
once daily	25.8% (77)
twice daily	55.7% (166)
≥twice daily	13.8% (41)
**Interproximal Cleaning Frequency**	
never	37.6% (112)
1–3 times/month	15.4% (46)
1–3 times/week	21.8% (65)
daily	24.2% (72)
**Number of Missing Teeth**	
none	63.4% (189)
1–2 teeth	24.3% (74)
≥3 teeth	11.4% (34)
**Ever Told Had Gum Disease**	
yes	10.4% (31)
**Dental Care When Not Pregnant**	
never	10.1% (30)
for pain or problem only	28.9% (86)
≤once yearly	45.9% (137)
≥twice yearly	14.8% (44)
**Usual Place of Routine Dental Care**	
no routine care	29.2% (87)
private dental office	43.0% (128)
public dental clinic	26.5% (79)
**Difficulty Finding Place for Routine Dental Care**	
not tried to find place for routine care	12.4% (37)
not difficult	44.3% (132)
somewhat difficult	23.2% (69)
very difficult	18.8% (56)
**Method of Payment for Dental Care when not Pregnant**	
private dental insurance	13.1% (39)
Medicaid	54.0% (161)
cash/self-pay	19.8% (59)
could not pay/don’t know how I would pay	9.4% (28)
**Oral Symptoms during Current Pregnancy**	
bleeding gums	47.7% (142)
oral/dental pain	41.6% (124)
swollen gums	31.2% (93)
cavity	28.5% (85)
loose tooth/teeth	10.1% (30)
**Dental Care Referral from Prenatal Clinic during Current Pregnancy**	72.5% (216)
**Dental Visit during Current Pregnancy**	56.0% (167)

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
