# Peer review of "A Successful Dental Care Referral Program for Low-Income Pregnant Women in New York"

_ijerph, 2021, doi:10.3390/ijerph182312724_

Round 1

Reviewer 1 Report

The final purpose of the work, reducing disparities and improving the oral health of pregnant women, has prevailed in our review, over some personal considerations and suggestions. Among them, in formulating the objective "to evaluate oral health ..." the subjectivity derived from the methodology used in its achievement (questionnaire), including, for example the term "perceived", avoideing the misunderstanding of the parameter professional evaluation. In this way, it would be interesting to contrast the data reported by the patients regarding their oral health with the clinical records made by professionals.

As it is a subject highly influenced by social variables, it would improve the discussion of the results obtained, their contrast with similar studies in populations from other countries, and also the updating of data from other American populations. We also miss an article published on the subject and from the same publication (year 2019).

The conclusions should be more closely related to the objectives set.

The bibliographic review, in line with what is referred to in the discussion, could be expanded with studies in other US populations and populations from other countries. The bibliographic needs to be enriched.

Author Response

Response to Reviewers

We wish to thank the reviewers for their thoughtful comments on our manuscript, “A Successful Dental Care Referral Program for Low-Income Pregnant Women in New York,” and for the opportunity to respond. We have organized our responses by issue raised in the table below and indicate where in the manuscript we have made changes that address each critique/suggestion.

reviewer

comment #

critique/suggestion

author response

location (line)

1

1

In formulating the objective "to evaluate oral health ..." the subjectivity derived from the methodology used in its achievement (questionnaire), including, for example the term "perceived", avoiding the misunderstanding of the parameter professional evaluation.

Oral health (perceived or otherwise) is not the primary focus of this study. We recognize that mentioning “oral health” in the abstract was confusing and have removed “oral health” from the abstract. We have also removed it from our study objectives as evaluation of self-reported oral health was not one of our primary study objectives. We describe self-rated oral health in the results to describe our population only.

We have also added the descriptors “self-reported” or “perceived” before “oral health” and before other variables (including “dental referral” and “dental care utilization”) to emphasize that all data were self-reported by the participants and not objectively measured by the investigators.

abstract (12)

introduction (58-59, 60)

results (100, 130)

Table 2

1

2

In this way, it would be interesting to contrast the data reported by the patients regarding their oral health with the clinical records made by professionals.

While it would be interesting to evaluate the correlation between subjective (self-reported) oral health and oral health as measured by a clinician/researcher, clinical examinations were beyond the scope of this study. We have specified that oral health was measured by self-report and there is “perceived oral health.”

1

3

As it is a subject highly influenced by social variables, it would improve the discussion of the results obtained, their contrast with similar studies in populations from other countries, and also the updating of data from other American populations.

We agree that a comparison of this program to other programs is warranted. To date there are only 4 studies that have evaluated rates of dental care utilization among their populations. These studies were all conducted in low-income populations, and all but one were US studies. We have added a paragraph that describes these studies in detail and compares our findings to those achieved elsewhere.

discussion (lines 187-208)

references added (21-25)

1

4

We also miss an article published on the subject and from the same publication (year 2019).

Please see response to comment #3, above

1

5

The conclusions should be more closely related to the objectives set.

We have edited our conclusions to mirror our two objectives and have edited our conclusion section.

conclusion (lines 247-252)

1

6

The bibliographic review, in line with what is referred to in the discussion, could be expanded with studies in other US populations and populations from other countries. The bibliographic needs to be enriched.

We have expanded our bibliography to reflect our expanded discussion of other programs in the US and abroad that were designed to increase dental care utilization among low-income pregnant women.

references (added 21-25)

2

1

In the introduction, give more space to the correlation between oral health and pregnancy

We have added several sentences that describe oral changes during pregnancy.

Introduction (lines 36-54)

2

2

When referring to a study or a protocol, enter the reference bibliographic entry and the results obtained from the research.

We have updated our manuscript to include all references and describe results from all studies mentioned in our text.

updated references

2

3

When it comes to medical insurance, briefly specify the type of health care existing in the reference place to contextualize the data collected.

We have added the sentence “Women seen in these facilities are low-income and are covered by Medicaid during pregnancy” in the Methods section in order to provide context.

methods

(lines 69-70)

2

4

specify who conducts the study

We have specified who conducted the study.

methods

(lines 77-78)

2

5

I would add perceived oral health being a self-administered questionnaire

We have specified that oral health was perceived by the respondents and have emphasized that this was a self-administered questionnaire. (see response to review #1, comment #1)

abstract (12)

introduction (58-59, 60)

results (100, 130)

Table 2

2

6

number the objectives correctly and can

also be classified as primary and secondary.

We have reduced our objectives to two primary objectives. Our introduction now states: “Our objectives were: 1) to evaluate whether women reported being referred for needed dental evaluation, prevention and care; 2) to determine whether those who reported referral were more likely to report having seen a dentist during pregnancy compared to those who did not report a referral.”

introduction (lines 80-82)

Again, thank you for this opportunity to respond. We have done our best to fully address key reviewer comments and incorporate feedback but would be happy to make any further revisions deemed necessary.

Reviewer 2 Report

the topic examined by the authors is of great interest and value, however the work needs a revision in its structuring and in its presentation of the data.
1- in the introduction, give more space to the correlation between oral health and pregnancy; when referring to a study or a protocol, enter the reference bibliographic entry and the results obtained from the research; when it comes to medical insurance, briefly specify the type of health care existing in the reference place to contextualize the data collected; specify who conducts the study; objective number 1 I would exclude dental medical treatment because it does not count in the article but; I would add perceived oral health being a self-administered questionnaire; number the objectives correctly and can also be classified as primary and secondary.
2- in the M&M section specify that the questionnaire is self-administered (if it is) and that the consent to the analysis of the data is present or if it is anonymous; specify the criteria with which test participants were selected (inclusion and exclusion criteria) and whether a sample analysis was conducted before defining the number of participants; specify what kind of criteria was used to select antenatal clinics; create subsections with -Sample size calculation - participants and data collection- statistical analysis-
3- in the results section line 133 leave only the percentage; bivariate analysis add one or more tables with the results; line 132 reported is repeated twice.
4 in the discussion and conclusions section many sentences are repeated

the section discusses and conclusions is repetitive in some sections.

it is advisable to follow the PRISMA CHECK LIST for the structuring of the study

Author Response

Response to Reviewers

We wish to thank the reviewers for their thoughtful comments on our manuscript, “A Successful Dental Care Referral Program for Low-Income Pregnant Women in New York,” and for the opportunity to respond. We have organized our responses by issue raised in the table below and indicate where in the manuscript we have made changes that address each critique/suggestion.

reviewer

comment #

critique/suggestion

author response

location (line)

1

1

In formulating the objective "to evaluate oral health ..." the subjectivity derived from the methodology used in its achievement (questionnaire), including, for example the term "perceived", avoiding the misunderstanding of the parameter professional evaluation.

Oral health (perceived or otherwise) is not the primary focus of this study. We recognize that mentioning “oral health” in the abstract was confusing and have removed “oral health” from the abstract. We have also removed it from our study objectives as evaluation of self-reported oral health was not one of our primary study objectives. We describe self-rated oral health in the results to describe our population only.

We have also added the descriptors “self-reported” or “perceived” before “oral health” and before other variables (including “dental referral” and “dental care utilization”) to emphasize that all data were self-reported by the participants and not objectively measured by the investigators.

abstract (12)

introduction (58-59, 60)

results (100, 130)

Table 2

1

2

In this way, it would be interesting to contrast the data reported by the patients regarding their oral health with the clinical records made by professionals.

While it would be interesting to evaluate the correlation between subjective (self-reported) oral health and oral health as measured by a clinician/researcher, clinical examinations were beyond the scope of this study. We have specified that oral health was measured by self-report and there is “perceived oral health.”

1

3

As it is a subject highly influenced by social variables, it would improve the discussion of the results obtained, their contrast with similar studies in populations from other countries, and also the updating of data from other American populations.

We agree that a comparison of this program to other programs is warranted. To date there are only 4 studies that have evaluated rates of dental care utilization among their populations. These studies were all conducted in low-income populations, and all but one were US studies. We have added a paragraph that describes these studies in detail and compares our findings to those achieved elsewhere.

discussion (lines 187-208)

references added (21-25)

1

4

We also miss an article published on the subject and from the same publication (year 2019).

Please see response to comment #3, above

1

5

The conclusions should be more closely related to the objectives set.

We have edited our conclusions to mirror our two objectives and have edited our conclusion section.

conclusion (lines 247-252)

1

6

The bibliographic review, in line with what is referred to in the discussion, could be expanded with studies in other US populations and populations from other countries. The bibliographic needs to be enriched.

We have expanded our bibliography to reflect our expanded discussion of other programs in the US and abroad that were designed to increase dental care utilization among low-income pregnant women.

references (added 21-25)

2

1

In the introduction, give more space to the correlation between oral health and pregnancy

We have added several sentences that describe oral changes during pregnancy.

Introduction (lines 36-54)

2

2

When referring to a study or a protocol, enter the reference bibliographic entry and the results obtained from the research.

We have updated our manuscript to include all references and describe results from all studies mentioned in our text.

updated references

2

3

When it comes to medical insurance, briefly specify the type of health care existing in the reference place to contextualize the data collected.

We have added the sentence “Women seen in these facilities are low-income and are covered by Medicaid during pregnancy” in the Methods section in order to provide context.

methods

(lines 69-70)

2

4

specify who conducts the study

We have specified who conducted the study.

methods

(lines 77-78)

2

5

I would add perceived oral health being a self-administered questionnaire

We have specified that oral health was perceived by the respondents and have emphasized that this was a self-administered questionnaire. (see response to review #1, comment #1)

abstract (12)

introduction (58-59, 60)

results (100, 130)

Table 2

2

6

number the objectives correctly and can

also be classified as primary and secondary.

We have reduced our objectives to two primary objectives. Our introduction now states: “Our objectives were: 1) to evaluate whether women reported being referred for needed dental evaluation, prevention and care; 2) to determine whether those who reported referral were more likely to report having seen a dentist during pregnancy compared to those who did not report a referral.”

introduction (lines 80-82)

Again, thank you for this opportunity to respond. We have done our best to fully address key reviewer comments and incorporate feedback but would be happy to make any further revisions deemed necessary.

This manuscript is a resubmission of an earlier submission. The following is a list of the peer review reports and author responses from that submission.